psychology

probabilistic reasoning, developmental psychology, intuitive statistics, child cognition

**Author for correspondence:**
Sarah Placì
e-mail: sarah.placi@gmail.com

†Equal contribution.

# Do infants and preschoolers quantify probabilities based on proportions?

Sarah Placì[1,2,3], Julia Fischer[1,2,†] and Hannes Rakoczy[2,3,†]

[1]Cognitive Ethology Laboratory, and [2]Leibniz ScienceCampus Primate Cognition, German Primate Center, Kellnerweg 4, 37077 Göttingen, Germany
[3]Department of Developmental Psychology, University of Göttingen, Waldweg 26, 37073 Göttingen, Germany

SP, 0000-0002-9652-8250; JF, 0000-0002-5807-0074

Most statistical problems encountered throughout life require the ability to quantify probabilities based on proportions. Recent findings on the early ontogeny of this ability have been mixed: For example, when presented with jars containing preferred and less preferred items, 12-month-olds, but not 3- and 4-years-olds, seem to rely on the proportions of objects in the jars to predict the content of samples randomly drawn out of them. Given these contrasting findings, it remains unclear what the probabilistic reasoning abilities of young children are and how they develop. In our study, we addressed this question and tested, with identical methods across age groups and similar methods to previous studies, whether 12-month-olds and 3- and 4-years-olds rely on proportions of objects to estimate probabilities of random sampling events. Results revealed that neither infants nor preschoolers do. While preschoolers' performance is in line with previous findings, infants' performance is difficult to interpret given their failure in a control condition in which the outcomes happened with certainty rather than a graded probability. More systematic studies are needed to explain why infants succeeded in a previous study but failed in our study.

## 1. Introduction

Probability judgements are an important component of rationality as they help people learn and make decisions under uncertainty. An increasing number of findings suggest that young children have intuitions about probabilities, which they use to make predictions about outcomes of random events [1–3]. These intuitions seem, under some conditions, to help children make rational decisions under uncertainty [4], and to be independent of schooling and culture [5]. However,

within these findings, there are inconsistencies in how children quantify probabilities and how probability intuitions develop during childhood. In particular, it is not clear whether young children can quantify probabilities based on proportions, an ability necessary to most probability problems [6].

In a study with 12-month-olds, for example, infants were first shown a pink and a black lollipop [4]. The experimenter placed both lollipops on the ground and waited for the infant to crawl towards one of them. The item chosen by the child was considered as her preferred item and the other item as the less preferred one. The experimenter then presented two transparent jars containing a mixture of both types of lollipops to the child. In one jar, there was a higher proportion of preferred lollipops (favourable jar) than in the other jar (unfavourable jar). The experimenter also brought two opaque cups that she placed on each side of the jars. She then concealed the content of both jars with two paper occluders, closed her eyes, reached into the jar on the right, grabbed a lollipop, and while holding it concealed in her fist so that only the stick was apparent, she moved it to one of the two cups. She repeated the same action with the jar on the left. The experimenter then encouraged the child to come to choose one of the two cups. Several conditions were tested: in some conditions, the absolute quantity of preferred lollipops was equal in both jars, and in others, it was higher in the unfavourable jar. For example, in Experiment 2, there were 16 preferred and 4 less preferred lollipops in the favourable jar, and 24 preferred and 96 less preferred lollipops in the unfavourable jar. This was meant to rule out that children solved the task by comparing absolute quantities of preferred items in both jars instead of estimating probabilities based on proportions. Results revealed that more children than expected by chance moved towards the cup containing the sample drawn out of the favourable jar and suggest that infants rely on proportions of large quantities of objects to make inferences about random drawing.

In three similar tasks [7], one of which was tested with 3-, 4- and 5-year-olds (Study 1), and the two others only with 3-years-olds (Studies 2 and 3), 3- and 4-years-olds consistently failed at quantifying probabilities based on proportions. Currently, we do not know why 12-month-olds performed better than 3- and 4-years-olds. The main differences between the lollipop study and Study 2 of the preschooler study—whose methods were closest to those of the lollipop study—were (i) the use of lollipops in one case and of spoons with stickers in the other, (ii) the age difference between subjects of both studies, and (iii) the fact that in the lollipop study, parents were not blind to the experiment.

One possible explanation for these differences in performance could be that the older children were not motivated enough by the stickers and did therefore not care enough to solve the task. Another explanation could be that the younger children relied on an alternative strategy to solve the task. For example, infants could have believed that there were always more preferred lollipops in the favourable jar and could have solved the task by comparing absolute quantities of visible preferred items. Consider Experiment 2 again. In the unfavourable jar, there were more preferred lollipops than in the favourable jar. Possibly, a substantial number of the preferred lollipops in the unfavourable jar ($n = 24$) were hidden by the high quantity of less preferred lollipops ($n = 96$), so that in appearance, more preferred lollipops were seen in the favourable jar. A third explanation could be that infants, instead of solving the task by estimating probabilities, simply responded to slight cues given by their parents who were not blind to the conditions and who were holding their child before the child's decision.

The findings of the infant study encouraged a series of comparative studies with other animals [8–13]. These studies are important to better apprehend the distribution of probabilistic reasoning skills and come to a better understanding of their evolution. However, given the uncertainty that still revolves around young children's probability intuitions, it might be premature to use the lollipop study as an established paradigm, and previous findings for children as established facts. To shed more light on the evolution and the development of probabilistic reasoning abilities, it seems therefore warranted to try and resolve the inconsistencies found in the child literature. That is the aim of the current study. More specifically, we aimed to test young children's ability to quantify probabilities based on proportions across two age groups, using the same methods. We tested 12-month-olds and 3- and 4-years-olds in a paradigm similar to the one used previously [4,7], to which we made some changes. In our study, the preferred and less preferred items were not stacked on top of each other in narrow containers. Instead, we selected large enough boxes that allowed good visibility on all items and therefore prevented children from solving the task by comparing absolute quantities of objects. We also changed the nature of the objects to make them more interesting as potential rewards. Finally, in our study, parents were blind to all conditions.

# 2. Methods

## 2.1. Participants

Sixty-eight children were included in the final sample, which was split into two age groups: infants ($n = 34$, mean age = 12 months, age range = 10–14 months) and preschoolers ($n = 34$, mean age = 46 months, age range = 36–54 months). We had planned to keep a final sample of 34 infants, as this would allow us to replicate the effect reported in Experiment 1 (i.e. that 75% of the 24 tested infants chose the favourable cup) of the lollipop study [4] with a 90% chance. Thirty-five additional children (thirty-four infants and one preschooler) were tested but not kept in the final sample because they either did not make any choice in the preference test or the test condition ($n = 6$) or because they did not make the expected choice in the preference trial ($n = 29$). Children were recruited from a database of families who had voluntarily registered and agreed to participate. They were from mixed socio-economic backgrounds and some families were plurilingual.

## 2.2. Materials

### 2.2.1. Objects

Transparent white and transparent blue Kinder eggs were used as stimuli for the experiment. To all eggs were attached thin wooden sticks of approximately 20 cm length. This measure was taken to allow children to see that only one object was removed from each box and transferred to a cup. On all blue eggs were also stuck white cartoon eyes to make them more interesting. Moreover, all blue eggs contained finger puppets representing different animals whereas all of the white eggs were empty (see electronic supplementary material, figure S1).

### 2.2.2. Boxes

Two cubical transparent boxes (length × width × height: 37.8 × 39.6 × 18.5 cm) were used as containers. Two white rectangular blankets were used to cover the boxes.

### 2.2.3. Cups

Two opaque cups (length × width × height: 10 × 10 × 15 cm) were used as containers for the eggs sampled from the boxes. Each cup was covered with a black cloth with a hole in the centre through which the experimenter introduced the eggs.

## 2.3. Procedure and design

Each child was tested individually in a quiet room, in the presence of a family member. Children were told they would play a game. They sat on the lap of one parent (except for some 3- and 4-years-olds who preferred to sit alone, with the parent sitting next to them) who sat on a chair in front of a table. The experimenter sat on the opposite side of the table. The two cups were already standing on the two extremities of the table, on each side of the child and the experimenter. Parents were asked to wear opaque glasses that prevented them from seeing the experiment. They were allowed to withdraw the glasses between each phase of the experiment. All children would first undergo a preference test, followed by a Probability condition and a Baseline condition (within-subject design).

## 2.4. Preference trial

The experimenter brought from under the table a transparent white egg and, drawing the attention of the child to it, said 'Look here, I have a white egg, I open it. Oh, it is unfortunately empty. See, there is nothing inside'. The experimenter closed the egg, brought a blue egg from under the table, and said 'Look here, I also have a blue egg, I open it. Oh, there is something inside'. She closed the blue egg again. Then, always starting with the white egg, she said 'Look here' and slowly hid it in one of the cups, then repeated the same with the blue egg. She finally lifted both cups simultaneously and presented them at an equal distance from the child who could try and grab the one she preferred. If the child chose the cup containing the blue egg, the experimenter said: 'Oh great, you found the blue

**Figure 1.** Procedure. The experimenter presented both boxes used in the Probability condition, one after the other, as depicted in (*a*). She then covered both boxes with white blankets and while closing her eyes, removed one object from the right box and hid it in the green cup adjacent to the box, as shown in (*b*). The same was repeated with the second box. She then took both cups, approached them at equal distance from the child and encouraged the child to choose a cup. The same procedure was repeated with the Baseline condition.

egg'. She opened the egg and allowed the child to play with the finger puppet for some time. If the child chose the cup containing the white egg the experimenter said: 'Oh, this egg is unfortunately empty'. Only children who chose the cup containing the blue egg were kept in the final sample.

## 2.5. Probability condition

The experimenter brought the two covered boxes on the table. One box contained nine blue eggs and three white eggs (favourable population), the other box contained 16 blue eggs and 48 white eggs (unfavourable population). The experimenter uncovered the first box, always starting with the one on the right (see figure 1). She said 'Look what is in this box' and moved the box towards the child to allow the child to see the content of the box. The experimenter shook the box to make clear that the eggs could be mixed. She repeated the same action with the second box. She then covered the two boxes again and, with her eyes closed, reached in the box on the right, hid one egg in her hand, and placed it in the cup adjacent to the box. She repeated the same action with the second box. She finally lifted both cups simultaneously and presented them at equal distance from the child, saying: 'pick the one you want' and waited for the child to try and reach for one cup. If the child chose the cup containing the blue egg the experimenter said: 'Oh great, you found the blue egg'. She opened the egg and allowed the child to play with the finger puppet for some time. If the child chose the cup containing the white egg, the experimenter said: 'Oh, this egg is unfortunately empty'.

## 2.6. Baseline condition

The same procedure as in the Probability condition was repeated, only this time, one box contained 48 blue eggs (favourable population) and the other box 48 white eggs (unfavourable population). This condition was added to check whether children understood the procedure. If children failed in this condition, the reason could not be a lack of probability intuitions, as the outcomes of the drawing

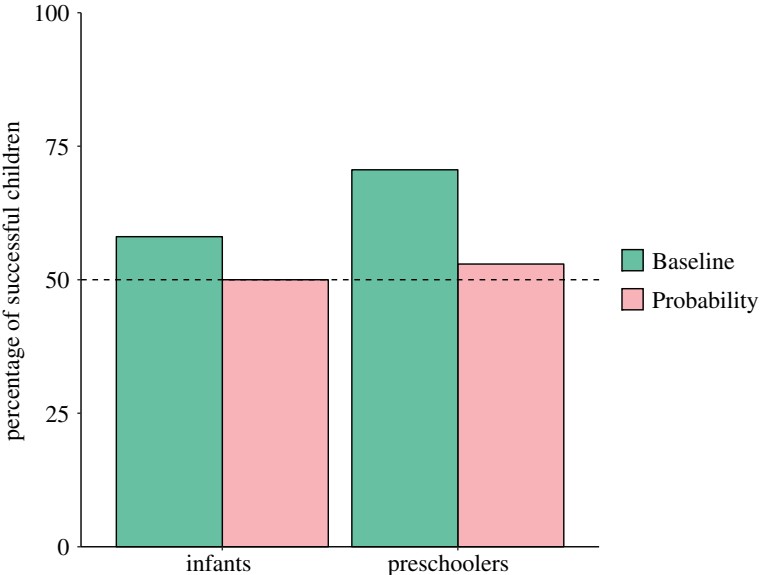

**Figure 2.** Percentage of successful children in each condition and each age category.

were certain. The side of the favourable sample and of the favourable population was counterbalanced between the preference trial, the Probability condition, and the Baseline condition.

## 2.7. Coding

Every session was video recorded. Children's choices were coded from the videos. Whenever children chose the sample stemming from the favourable population, we considered it as a success, and as a failure when they chose the alternative option.

## 2.8. Predictions

In the Probability condition, we predicted that if children rely on a comparison of absolute quantities of preferred items to make predictions, they would choose the sample drawn out of the unfavourable population at above chance level. At this age, they can discriminate 2 : 3 ratios [14] and should, therefore, be able to distinguish that there are more blue eggs in the unfavourable population. If they rely on proportions to quantify probabilities, we predicted that children would choose the sample drawn out of the favourable population at above chance level.

## 3. Results

In the Baseline condition, 18 out of 31 (58%) infants selected the cup with the sample drawn out of the favourable population, no different from chance (binomial test, $p = 0.47$; 95% confidence interval [39,75]). Twenty-four out of 34 (71%) preschoolers selected the cup with the sample drawn out of the favourable population, significantly different from chance (binomial test, $p = 0.02$; 95% confidence interval [53,85], see figure 2).

In the Probability condition, 17 out of 34 (50%) infants selected the cup with the sample drawn out of the favourable population, no different from chance (binomial test, $p = 1$; 95% confidence interval [32,68]). Eighteen out of 34 (53%) preschoolers selected the cup with the sample drawn out of the favourable population, no different from chance (binomial test, $p = 0.86$; 95% confidence interval [35,70], see figure 2).

Among the preschoolers who chose the correct sample in the Baseline condition, 58% (14 out of 24) also succeeded in the Probability condition, no different from chance (binomial test, $p = 0.54$; 95% confidence interval [37,78]).

To test whether age had an effect on infants' and preschoolers' performance, we did, for each condition and each age category, a *post hoc* logistic regression, with subjects' choice as response variable and age (in months) as predictor. For this, we used the glm function of the R package lme4

(R Core Team 2017). Results showed that age had no effect in any age category and any condition (infants in Probability condition: estimate ± s.e. = 0.39 ± 0.25, $z = 1.52$, $p = 0.13$; preschoolers in Probability condition: estimate ± s.e. = −0.05 ± 0.07, $z = −0.70$, $p = 0.49$; infants in Baseline condition: estimate ± s.e. = 0.23 ± 0.27, $z = 0.85$, $p = 0.39$; preschoolers in Baseline condition: estimate ± s.e. = −0.05 ± 0.08, $z = −0.70$, $p = 0.48$).

## 4. Discussion

In summary, we found that in our study, neither 12-month-olds nor 3- and 4-years-olds selected the favourable sample above chance level in the Probability condition, and only 3- and 4-years-olds did so in the Baseline condition. These results suggest that preschoolers did not rely on proportions of objects in the boxes to make inferences about the samples in the Probability condition. We can rule out that their failure was due to the procedure, as in the Baseline condition they selected the correct cup at above chance level. These results replicate previous results [7]. It is nonetheless intriguing that only 71% of preschoolers selected the correct cup in the Baseline condition. This cannot be attributed to a weak preference for the blue eggs, as only one preschooler out of 35 did not choose the cup containing the blue egg in the preference test (and was therefore not included in the final sample). It seems that something in the procedure confused some children. This confusing factor could have been the experimenter performing the drawing: some children could have expected her to play a trick on them and did therefore not consider the outcome of the drawing to be certain. In a different study, but a condition similar to our Baseline condition ([7], Study 3), the drawing was performed by a machine, and children's performance was much better (90% success). However, note that even then, they performed at chance level when they had to reason about proportions [7].

One could wonder whether in our study there was a subsample of preschoolers who were not confused by the procedure and who performed well in the Probability condition. This seems not to have been the case, as only 58% of the preschoolers who performed well in the Baseline condition also performed well in the Probability condition.

Our findings also indicate that infants did not rely on the proportions of objects in the boxes to make inferences about the samples in the Probability condition (i.e. did not preferentially choose the sample drawn out of the favourable box). However, this is not surprising given that infants seem not to have relied on any numerical information at all, as indicated by their at-chance performance in the Baseline condition. These null results are unlikely to be due to low power, as we planned our sample size so that it would allow us to find the effect reported in the previous study with infants [4] with a 90% chance.

A possible explanation for infants' failure in both conditions is that they did not have a strong preference for the blue eggs. Almost half of the infants tested in the preference trial (28 infants out of 62; 45%) did not select the cup containing the blue egg and were therefore excluded from the final sample. Maybe, rather than having selected a sample of children with a preference for the blue eggs, we selected a sample of children who chose randomly between both egg colours. It is hard to explain why infants would not have a strong preference for an egg with cartoon eyes that visibly contains something when the alternative is an empty egg without eyes. Perhaps, the fact that the objects did or did not contain other objects was confusing to them. In the lollipop study [4], it is not clear whether infants had a strong preference for any lollipop. The only indicator for such preference is that, in the test trial, a significant proportion of children seemed to try and find the same lollipop they chose in the preference test. However, children's choices in the test trial could also have been motivated by alternative factors (e.g. the influence of parents, see below) and therefore be no indicator of any preference. With only one preference trial, it is impossible to say whether any particular child preferred an object, or whether she chose randomly between two alternatives.

There were differences between the previous lollipop study [4] and the present one in several other respects. Regarding preference trials, we placed the objects in the cups instead of placing them visibly in front of the children. Children, therefore, had to keep track of the location of both eggs. However, as infants are able to retrieve hidden objects from eight-months onward [15] and to find different objects in different locations [16], this should not explain why only 55% of the infants in our preference trial chose the cup containing the blue egg. Regarding the target-dependent variable, we measured children's decisions indicated in their reaching behaviour (i.e. which cup they tried to reach for), whereas the lollipop study [4] measured children's decision in terms of the cup towards which children moved. In the present study, in line with previous studies on non-human primates [8–12], we chose a reaching task because it was difficult for parents wearing opaque glasses to hold their

children on the floor. However, it seems unlikely that preferential reaching tasks should be more difficult than differential crawling/movement tasks (see [17], in which infants perform well in a reaching task). Finally, we decided to use large enough boxes instead of narrower jars to allow children to see their entire content. In terms of probabilities, this decision should also have created a better impression of a random drawing, as all items in the boxes were equally reachable, which might not have been the case if items were all stacked on top of each other, as was the case in the lollipop study [4]. As the boxes were larger (length × width × height: 37.8 × 39.6 × 18.5 cm) and their content spread out, we made sure that children could see the content of each box by bringing each box closer to them and attracting their attention before any object was sampled. One could speculate whether it might have been more difficult for children to gaze at the contents of both boxes simultaneously to compare their populations. However, there is no reason to believe that children must see proportions simultaneously to compare them, as this is not the case when they compare absolute quantities [14].

Another difference between our experiment and the experiments in the lollipop study [4] is, as mentioned above, that in our case, parents could not have influenced their child's behaviour, as they were blind to the experiment. In the lollipop study [4], parents were not blind and were holding their children before the children's decisions. Even if parents were asked not to interfere with the task, they might have done so inadvertently and in this way produced positive results. It is therefore not possible to say why infants performed well in the lollipop study [4]. Was it a real demonstration of their probability intuitions or was it because they received help from their parents? It is also not possible to say why infants performed poorly in our study. Was it because they did not have any preference for any object or because reasoning about sampling events is too hard for them? Preschoolers' repeated failure to assess probabilities based on proportions, which has now been documented across several experiments and studies (three experiments in [7] and one experiment in the present study), casts additional doubt on infants' performance in the lollipop study [4]. It is hard to believe that infants are good at tasks at which preschoolers fail. Nonetheless, more research is necessary to address these questions. Future research should assess children's preferences with several preference trials and should make sure that parents cannot interfere with their child's decisions. It is also worth asking whether this paradigm does need any other further improvement, considering the all-but-perfect performance of preschoolers in the Baseline condition. There seems to be something in the procedure that adds noise to the data independently of children's ability to estimate probabilities.

One could wonder why young children might fail at tasks at which other non-human animals seem to succeed. Non-human great apes [8–10], capuchin monkeys [11], one long-tailed macaque [12] and keas [13] were reported to succeed in tasks in which they had to make probability estimations and decisions based on proportions. Possibly, the methods used in these comparative studies were more suitable to uncover competence in non-human animals than in human children, i.e. the goal of the task was maybe more obvious to non-human animals than to children. As mentioned above, 3- and 4-years-olds might have tried to reach alternative goals, e.g. not being tricked, which might have interfered with the more obvious goal of choosing the favourable sample. However, this seems unlikely, as 3-years-olds also failed under different conditions that did not involve a human experimenter performing the drawing [7]. It is also possible that humans and some non-human animals share the same capacity, but that, contrary to what has been reported in previous studies [4], this capacity only develops around 5 years of age in humans—this possibility is not unheard of in other areas of cognition (see [18]). Finally, it is also possible that neither young children nor any other tested animal quantified probabilities based on proportions. In previous studies, non-human animals could have based their inferences on the absolute number of preferred items that were visible (which we discussed in the Introduction), rather than on the proportions of preferred items, especially given that these items were food, which is a highly salient stimulus that might deflect reasoning in decision-making tasks [19–21]. This alternative strategy has only been explored in one study with long-tailed macaques [12] and could only be eliminated for one individual. Subjects could also have relied on a combination of heuristics, a strategy that children up to 10 years of age seem to use when tested with populations containing fewer elements [22]. In the light of these considerations, a better question might be whether we have at the moment, strong enough evidence to conclude that children younger than 5 years of age and non-human animals can quantify probabilities based on proportions and whether the methods that were used to investigate the presence of this ability can detect it.

In summary, the present study failed to find evidence that 3- and 4-years-olds rely on proportions of objects to make inferences about sampling events. These findings are congruent with previous results [7]. Contrary to previous results [4], the present study also failed to find evidence for intuitive probabilistic inferences based on proportions in infants, but these findings remain difficult to interpret in light of

infants' failure in the non-probabilistic Baseline condition. However, previous findings with infants are also hard to interpret given that parents could have influenced their child's decisions. In general, there seems to be uncertainty as to the validity of this paradigm. This should, in the future, motivate researchers to work with alternative paradigms or improved ones and to try to replicate previous findings with children, but also with non-human primates, if we want to come to a better understanding of the evolution and development of probabilistic reasoning abilities.

Ethics. This research was conducted in accordance with the Declaration of Helsinki and the Ethical Principles of the German Psychological Society (DGPs), the Association of German Professional Psychologists (BDP), and the American Psychological Association (APA). It involved no invasive or otherwise ethically problematic techniques and no deception (and therefore, according to National jurisdiction, did not require a separate vote by a local Institutional Review Board; see the regulations on freedom of research in the German Constitution (§ 5 (3)), and the German University Law (§ 22)). Before the study started, informed consent was obtained from the parents of the subjects.

Data accessibility. Supporting material has been uploaded as part of the electronic supplementary material. The datasets generated and analysed during the current study, as well as the R script used for the analyses, have been deposited at the Dryad Digital Repository and can be accessed at https://doi.org/10.5061/dryad.0gb5mkkwp.

Authors' contributions. All authors contributed to the conception and the design of the study, S.P. collected, analysed and interpreted the data, and drafted the manuscript. J.F. and H.R. commented on the manuscript and edited the text.

Competing interests. The authors declare no competing interests.

Funding. This work is supported by the Deutsche Forschungsgemeinschaft (DFG, German Research Foundation – Project number 254142454/GRK 207).

Acknowledgements. We thank Isabel Ganter and Lydia Schidelko for their help with the data collection, and all parents and children who participated in our study.

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
