## [Reviewer comments · Royal Society Open Science]

Review History

RSOS-191751.R0 (Original submission)

Review form: Reviewer 1

Is the manuscript scientifically sound in its present form?

Yes

Are the interpretations and conclusions justified by the results?

Yes

Is the language acceptable?

Yes

Do you have any ethical concerns with this paper?

No

Have you any concerns about statistical analyses in this paper?

No

Recommendation?

Accept with minor revision (please list in comments)

Comments to the Author(s)

I believe this is a very interesting paper, reporting well-conducted experiments with properly-analysed data. I recommend publication. I have only a couple of minor comments.

- On line 41 consider mentioning that these capabilities can be observed regardless of schooling and culture (e.g. Fontanari et al PNAS December 2, 2014 111 (48) 17075-17080).

- Line 231: perhaps better "by the fact"

Review form: Reviewer 2**Is the manuscript scientifically sound in its present form?**

Yes

Are the interpretations and conclusions justified by the results?

Yes

Is the language acceptable?

Yes

Do you have any ethical concerns with this paper?

No

Have you any concerns about statistical analyses in this paper?

No

Recommendation?

Reject

Comments to the Author(s)

I read the manuscript "Do infants and preschoolers quantify probabilities based on proportions".

The authors present the result of one experiment ran with 12-month-olds and 3- to 4-year-olds, aiming at reconciling some discrepancies in the literature. Some papers indeed reports infants' ability to compute probabilities, while other studies find that pre-schoolers fail to do so.

The manuscript is very clear and well written.

The only problem is that it is unclear what to make of the results. The authors find that infants fail both the probability test and the baseline test. I fully agree with what the authors write in the discussion. This non-results could be due to a methodological issue: infants do not exhibit a strong preference for the outcome that should be preferred. The pity is that the authors stop here. I believe additional experimental work is necessary before this work is published. Could the authors improve their methods to better assess the preference of infants? Maybe with at least 2 or 3 preference tests. Or using different objects. In the present test, there is nothing to conclude from the infant study.

The results are more interesting concerning the pre-school children. The authors find that children have (as a group) a consistent preference for one of the objects, succeed in the baseline trial, but fail to use probabilities. This replicates previous results in the literature. Now the problem is that it is still a negative result, and the authors rightly discuss the possibility that children would succeed if the methods are improved. Again, I would suggest that the authors do run studies with improved methods, in order to reach stronger conclusions, these being positive or negative.

In sum, the paper is well written and the topic is very interesting. The authors raise a number of methodological issue, but they should follow-up with experimental work, in order to make a useful contribution to the literature.

Review form: Reviewer 3

Is the manuscript scientifically sound in its present form?

No

Are the interpretations and conclusions justified by the results?

No

Is the language acceptable?

Yes

Do you have any ethical concerns with this paper?

No

Have you any concerns about statistical analyses in this paper?

No

Recommendation?

Major revision is needed (please make suggestions in comments)

Comments to the Author(s)

In the present experiment, preschoolers and infants completed a task in which they were required to make predictions about where to search for an object based on numerical information. First they were shown one “desirable” type of item and another less “desirable” type and their preference between those items was assessed. Then they were shown trials with either probabilistic distributions (ones in which proportions and absolute numbers conflicted) and deterministic distributions (uniform distributions of desirable versus undesirable items). Preschools performed above chance (though not particularly well) on the deterministic trial and both ages performed at chance in the probabilistic trial. Notably, infants did not appear to systematically prefer one type of item.

Comments:

(1) I generally agree with the authors’ discussions in the paper. It is likely that the paradigm did not work particularly well for infants, in that they were probably choosing at chance between the items. This makes their data quite difficult to interpret. The preschool data is more interpretable and does align with previous findings (Giroto et al., 2016). The baseline (deterministic) trial complicates the picture a little. I keep up on this literature actively and think a great deal about these paradigms. I agree with the authors that the hidden item search might add noise to the data and perhaps researchers in this area should consider alternative paradigms. I also agree that the preference aspect is tricky with infants. In Denison & Xu (2014), the preferable lollipop illuminates. It also takes really no time and involves no complications in demonstrating to the infant why it is preferable (unlike the finger puppets in the eggs – which I think was clever and I can see why the authors tried it). I wonder if the sparkles and lighting is a little more endogenously preferable (like the food in the non-human primate versions) than the association between the blue egg and the finger puppet.

In any case, I agree that there are more aspects to explore about this paradigm, or perhaps entirely different paradigms should be considered (anticipatory looking versions or something similar). On the whole, I’m not sure there’s quite enough here for a paper in this outlet, with the single experiment and the uncertainty in the infant results.

(2) If the paper were to be published, particularly in this journal, it seems a discussion of the success of non-human primates (and other species) in similar tasks and the failure here with preschoolers and (maybe) infants, should be added to the discussion.

(3) The spreading out of the stimuli so that the participants can see all of it is interesting. Again, I see why the authors did it (based on their rationale about the potential for items to be hidden in the previous infant work). But the figure makes me wonder if this would be difficult for the children to really see, because it is flat on the table. Another major difference, unless I'm mistaken, between this and the previous infant version is that this is a strictly reaching task and not a walking/crawling task (it wasn't stated explicitly but the stimuli are described as being on a table so I'm guessing it was a reaching task), which could be consequential.

Decision letter (RSOS-191751.R0)

28-Jan-2020

Dear Miss Placì,

The editors assigned to your paper ("Do infants and preschoolers quantify probabilities based on proportions?") have now received comments from reviewers. We would like you to revise your paper in accordance with the referee and Associate Editor suggestions which can be found below (not including confidential reports to the Editor). Please note this decision does not guarantee eventual acceptance.

Please submit a copy of your revised paper before 20-Feb-2020. Please note that the revision deadline will expire at 00.00am on this date. If we do not hear from you within this time then it will be assumed that the paper has been withdrawn. In exceptional circumstances, extensions may be possible if agreed with the Editorial Office in advance. We do not allow multiple rounds of revision so we urge you to make every effort to fully address all of the comments at this stage. If deemed necessary by the Editors, your manuscript will be sent back to one or more of the original reviewers for assessment. If the original reviewers are not available, we may invite new reviewers.

- Data accessibility

<http://datadryad.org/submit?journalID=RSOS&manu=RSOS-191751>

- Competing interests

- Authors' contributions

- Acknowledgements

- Funding statement

Kind regards,

Andrew Dunn

on behalf of Dr Denes Szucs (Associate Editor) and Essi Viding (Subject Editor)

Associate Editor's comments (Dr Denes Szucs):

Associate Editor: 1

Comments to the Author:

Please respond very thoroughly to Reviewers 2 and 3; please contextualize your results much more and explain your conclusions more; especially given that the current Introduction and Discussion are very short and cite only a few papers. Please also clarify whether your results may be due to low statistical power or not.

Comments to Author:

Reviewers' Comments to Author:

Reviewer: 1

Comments to the Author(s)

I believe this is a very interesting paper, reporting well-conducted experiments with properly-analysed data. I recommend publication. I have only a couple of minor comments.

- On line 41 consider mentioning that these capabilities can be observed regardless of schooling and culture (e.g. Fontanari et al PNAS December 2, 2014 111 (48) 17075-17080).

- Line 231: perhaps better "by the fact"

Reviewer: 2

Comments to the Author(s)

I read the manuscript "Do infants and preschoolers quantify probabilities based on proportions". The authors present the result of one experiment ran with 12-month-olds and 3- to 4-year-olds, aiming at reconciling some discrepancies in the literature. Some papers indeed reports infants' ability to compute probabilities, while other studies find that pre-schoolers fail to do so.

The manuscript is very clear and well written.

The only problem is that it is unclear what to make of the results. The authors find that infants fail both the probability test and the baseline test. I fully agree with what the authors write in the discussion. This non-results could be due to a methodological issue: infants do not exhibit a strong preference for the outcome that should be preferred. The pity is that the authors stop here. I believe additional experimental work is necessary before this work is published. Could the authors improve their methods to better assess the preference of infants? Maybe with at least 2 or 3 preference tests. Or using different objects. In the present test, there is nothing to conclude from the infant study.

The results are more interesting concerning the pre-school children. The authors find that children have (as a group) a consistent preference for one of the objects, succeed in the baseline trial, but fail to use probabilities. This replicates previous results in the literature. Now the problem is that it is still a negative result, and the authors rightly discuss the possibility that children would succeed if the methods are improved. Again, I would suggest that the authors do run studies with improved methods, in order to reach stronger conclusions, these being positive or negative.

In sum, the paper is well written and the topic is very interesting. The authors raise a number of methodological issue, but they should follow-up with experimental work, in order to make a useful contribution to the literature.

Reviewer: 3

Comments to the Author(s)

In the present experiment, preschoolers and infants completed a task in which they were required to make predictions about where to search for an object based on numerical information. First they were shown one "desirable" type of item and another less "desirable" type and their

preference between those items was assessed. Then they were shown trials with either probabilistic distributions (ones in which proportions and absolute numbers conflicted) and deterministic distributions (uniform distributions of desirable versus undesirable items). Preschools performed above chance (though not particularly well) on the deterministic trial and both ages performed at chance in the probabilistic trial. Notably, infants did not appear to systematically prefer one type of item.

Comments:

(1) I generally agree with the authors' discussions in the paper. It is likely that the paradigm did not work particularly well for infants, in that they were probably choosing at chance between the items. This makes their data quite difficult to interpret. The preschool data is more interpretable and does align with previous findings (Giroto et al., 2016). The baseline (deterministic) trial complicates the picture a little. I keep up on this literature actively and think a great deal about these paradigms. I agree with the authors that the hidden item search might add noise to the data and perhaps researchers in this area should consider alternative paradigms. I also agree that the preference aspect is tricky with infants. In Denison & Xu (2014), the preferable lollipop illuminates. It also takes really no time and involves no complications in demonstrating to the infant why it is preferable (unlike the finger puppets in the eggs - which I think was clever and I can see why the authors tried it). I wonder if the sparkles and lighting is a little more endogenously preferable (like the food in the non-human primate versions) than the association between the blue egg and the finger puppet.

In any case, I agree that there are more aspects to explore about this paradigm, or perhaps entirely different paradigms should be considered (anticipatory looking versions or something similar). On the whole, I'm not sure there's quite enough here for a paper in this outlet, with the single experiment and the uncertainty in the infant results.

(2) If the paper were to be published, particularly in this journal, it seems a discussion of the success of non-human primates (and other species) in similar tasks and the failure here with preschoolers and (maybe) infants, should be added to the discussion.

(3) The spreading out of the stimuli so that the participants can see all of it is interesting. Again, I see why the authors did it (based on their rationale about the potential for items to be hidden in the previous infant work). But the figure makes me wonder if this would be difficult for the children to really see, because it is flat on the table. Another major difference, unless I'm mistaken, between this and the previous infant version is that this is a strictly reaching task and not a walking/crawling task (it wasn't stated explicitly but the stimuli are described as being on a table so I'm guessing it was a reaching task), which could be consequential.

Author's Response to Decision Letter for (RSOS-191751.R0)

See Appendix A.

RSOS-191751.R1 (Revision)

Review form: Reviewer 2

Is the manuscript scientifically sound in its present form?

Yes

Are the interpretations and conclusions justified by the results?

Yes

Is the language acceptable?

Yes

Do you have any ethical concerns with this paper?

No

Have you any concerns about statistical analyses in this paper?

No

Recommendation?

Reject

Comments to the Author(s)

I read the new version of the manuscript "Do infants and preschoolers quantify probabilities based on proportions?".

In my previous review, I pointed that it is hard to conclude much from this work, despite the topic being interesting and the paper being well written. It is an experimental paper and the experimental results are not conclusive. The infant experiment does not give interpretable results. The authors discuss that the methods should be improved. I think the authors need to run such improved study to make a useful contribution.

The preschoolers' study gives interpretable results, but in light of the failure of infants, it is clear that the methods could be improved and that preschoolers may succeed with improved methods. In sum, I feel there are too many open questions for this paper to be published as it is. More experimental work is needed.

Decision letter (RSOS-191751.R1)

Dear Miss Placì,

On behalf of the Editors, I am pleased to inform you that your Manuscript RSOS-191751.R1 entitled "Do infants and preschoolers quantify probabilities based on proportions?" has been accepted for publication in Royal Society Open Science subject to minor revision in accordance with the referee suggestions. Please find the referees' comments at the end of this email.

Note that, on this occasion, there is a divergence of views between the reviewer and the Editor; however, as the latter make decisions based not only on the advice of referees but their own reading of the paper, they have opted to accept your paper on condition you attempt to respond to the remaining queries/comments they and the referee have.

The reviewers and Subject Editor have recommended publication, but also suggest some minor revisions to your manuscript. Therefore, I invite you to respond to the comments and revise your manuscript.

- Ethics statement

- Data accessibility

<http://datadryad.org/submit?journalID=RSOS&manu=RSOS-191751.R1>

- Competing interests

- Authors' contributions

- Acknowledgements

- Funding statement

Because the schedule for publication is very tight, it is a condition of publication that you submit the revised version of your manuscript before 06-Aug-2020. Please note that the revision

deadline will expire at 00.00am on this date. If you do not think you will be able to meet this date please let me know immediately.

on behalf of Dr Denes Szucs (Associate Editor)
openscience@royalsociety.org

Associate Editor Comments to Author (Dr Denes Szucs):

The use of English needs small corrections at some places.

The null finding in case of infants cannot be attributed to low statistical power. Hence, the study is actually interesting because of the inconclusive results: it casts doubt on an important paradigm used in the infant literature. Or, at least it suggests that the paradigm cannot be used as reliably as assumed. This may become an interesting study triggering debates in the literature.

Reviewer comments to Author:

Reviewer: 2

Comments to the Author(s)

I read the new version of the manuscript "Do infants and preschoolers quantify probabilities based on proportions?".

In my previous review, I pointed that it is hard to conclude much from this work, despite the topic being interesting and the paper being well written. It is an experimental paper and the experimental results are not conclusive. The infant experiment does not give interpretable results. The authors discuss that the methods should be improved. I think the authors need to run such improved study to make a useful contribution.

The preschoolers' study gives interpretable results, but in light of the failure of infants, it is clear that the methods could be improved and that preschoolers may succeed with improved methods. In sum, I feel there are too many open questions for this paper to be published as it is. More experimental work is needed.

Author's Response to Decision Letter for (RSOS-191751.R1)

See Appendix B.

Decision letter (RSOS-191751.R2)

Dear Miss Placi,

It is a pleasure to accept your manuscript entitled "Do infants and preschoolers quantify probabilities based on proportions?" in its current form for publication in Royal Society Open Science.

on behalf of Dr Denes Szucs (Associate Editor)
openscience@royalsociety.org

Appendix A

Associate Editor's comments (Dr Denes Szucs):

Associate Editor: 1

Comments to the Author:

Please respond very thoroughly to Reviewers 2 and 3; please contextualize your results much more and explain your conclusions more; especially given that the current Introduction and Discussion are very short and cite only a few papers. Please also clarify whether your results may be due to low statistical power or not.

RESPONSE: We contextualized our results much more and extended the introduction (l.87–92) and the discussion (l.256–325) to better explain our conclusions. We responded thoroughly to Reviewers 2 and 3. We argued that despite our inconclusive results with the infants, we still believe that our findings are of importance to the scientific community, because they raise awareness about the lack of strong evidence showing that children younger than 5 years of age can quantify probabilities based on proportions, and because they question the validity of a paradigm that seems on its way to be “established” as a cognitive task measuring probabilistic reasoning abilities (a new study with kea using similar methods has just been published!).

Moreover, our results cannot be due to low statistical power, because we pre-planned our sample size in such a way that it would allow us to replicate the effect reported in the previous study with infants with a 90% probability. We added this in the manuscript (l.105–108; l.239–241).

Reviewer #1

RESPONSE: We thank the Reviewer for the constructive comments. We did add the reference they suggested, it was indeed an important one.

Reviewer #2

RESPONSE: We thank the Reviewer for the constructive comments.

1. In sum, the paper is well written and the topic is very interesting. The authors raise a number of methodological issue, but they should follow-up with experimental work, in order to make a useful contribution to the literature.

RESPONSE: We do agree that our null results with infants do not contribute to a better understanding of their probabilistic abilities. However, we think our results are nonetheless of importance because they raise awareness about the lack of strong evidence that children younger than 5 years of age can reason about proportions, which in turn, raises questions about the performance of other animals that have been tested with a similar paradigm. To raise such concerns is crucial, especially when a paradigm seems to be on its way to become “established” as a valid task to assess the presence of some cognitive ability.

Reviewer #3

RESPONSE: We thank the Reviewer for the helpful and constructive comments. We hope we addressed all of the suggestions in a satisfying way and think the manuscript is much improved.

1. I generally agree with the authors' discussions in the paper. It is likely that the paradigm did not work particularly well for infants, in that they were probably choosing at chance between the items. This makes their data quite difficult to interpret. The preschool data is more interpretable and does align with previous findings (Giroto et al., 2016). The

baseline (deterministic) trial complicates the picture a little. I keep up on this literature actively and think a great deal about these paradigms. I agree with the authors that the hidden item search might add noise to the data and perhaps researchers in this area should consider alternative paradigms. I also agree that the preference aspect is tricky with infants. In Denison & Xu (2014), the preferable lollipop illuminates. It also takes really no time and involves no complications in demonstrating to the infant why it is preferable (unlike the finger puppets in the eggs – which I think was clever and I can see why the authors tried it). I wonder if the sparkles and lighting is a little more endogenously preferable (like the food in the non-human primate versions) than the association between the blue egg and the finger puppet.

In any case, I agree that there are more aspects to explore about this paradigm, or perhaps entirely different paradigms should be considered (anticipatory looking versions or something similar). On the whole, I'm not sure there's quite enough here for a paper in this outlet, with the single experiment and the uncertainty in the infant results.

RESPONSE: We do agree that in the experiments with infants, the preference aspect is tricky. We made this more salient in our discussion (l.247–255). However, note that in the first experiment of Denison & Xu (2014), the lollipops did not illuminate. Some were wrapped in black construction paper, others in pink, and the pink lollipops had in addition three golden stars on each side. The authors do not provide the proportion of children that chose the pink lollipop, but as in Experiment 2, they try to make the pink lollipops more appealing to increase children's preference, one might wonder whether they had any preference in Experiment 1 (in which 75% still chose the favorable cup in the test trial). The reason we think there is enough for a paper, is because our results do shed light about the uncertainty that revolves around this paradigm and the findings it produced, be it with human children or with nonhuman primates. It is important that researchers interested in probabilistic reasoning and comparative cognition are made aware of this.

2. If the paper were to be published, particularly in this journal, it seems a discussion of the success of non-human primates (and other species) in similar tasks and the failure here with preschoolers and (maybe) infants, should be added to the discussion.

RESPONSE: We thank the Reviewer for this suggestion. We added a discussion of the success of other animals (not only primates as we also mentioned a new study with kea) and the failure of preschoolers and (maybe) infants (l.297–325) and think the paper is much improved.

3. The spreading out of the stimuli so that the participants can see all of it is interesting. Again, I see why the authors did it (based on their rationale about the potential for items to be hidden in the previous infant work). But the figure makes me wonder if this would be difficult for the children to really see, because it is flat on the table. Another major difference, unless I'm mistaken, between this and the previous infant version is that this is a strictly reaching task and not a walking/crawling task (it wasn't stated explicitly but the stimuli are described as being on a table so I'm guessing it was a reaching task), which could be consequential.

RESPONSE: We thank the Reviewer for raising these additional points. We now discuss the possible implications of spreading the stimuli in larger boxes (l.269–281) and of using a reaching task (l.262–269) more thoroughly.

Appendix B

Associate Editor Comments to Author (Dr Denes Szucs):

The use of English needs small corrections at some places.

The null finding in case of infants cannot be attributed to low statistical power. Hence, the study is actually interesting because of the inconclusive results: it casts doubt on an important paradigm used in the infant literature. Or, at least it suggests that the paradigm cannot be used as reliably as assumed. This may become an interesting study triggering debates in the literature.

RESPONSE: We thank the associate editor for his comment. We went through the entire manuscript and corrected English errors and reformulated sentences where it seemed warranted. We also added additional information in the ethics statement (l.355-356), regarding parents' informed consent. Information on data accessibility and competing interest was already included in the manuscript.

Reviewer: 2

I read the new version of the manuscript "Do infants and preschoolers quantify probabilities based on proportions?".

In my previous review, I pointed that it is hard to conclude much from this work, despite the topic being interesting and the paper being well written. It is an experimental paper and the experimental results are not conclusive. The infant experiment does not give interpretable results. The authors discuss that the methods should be improved. I think the authors need to run such improved study to make a useful contribution.

The preschoolers' study gives interpretable results, but in light of the failure of infants, it is clear that the methods could be improved and that preschoolers may succeed with improved methods.

In sum, I feel there are too many open questions for this paper to be published as it is. More experimental work is needed.

RESPONSE: We thank the reviewer for her/his comment. We agree, once more, that it is hard to interpret our infant results. However, we think that our results raise awareness about the uncertainty that still revolves around the probabilistic reasoning abilities of infants in general. Furthermore, we do not believe that in our experiment the failure of infants can explain the failure of preschoolers. Preschoolers succeeded in the Baseline condition and their performance was comparable to the performance of preschoolers in Girotto et al. (2016), where different methods were used. Including ours, there are now four experiments (three in Girotto et al. (2016)) showing that preschoolers struggle with estimating probabilities based on proportions. This in turn casts even more doubt on the good performance of infants in Denison and Xu (2014). It is hard to believe that infants are good at tasks at which preschoolers fail. We reformulated some sentences to insist on these points (l.292-300). We agree that more work is needed to disentangle all these questions, and we encourage it. Nonetheless, we believe that our paper has the potential to raise the attention of the child cognition and the comparative cognition communities towards the fact that young children's ability to estimate probabilities based on proportion has not yet been convincingly established.